# Depressive Symptoms and Suicidal Ideation in Individuals Living Alone in South Korea

**DOI:** 10.3390/diagnostics12030603

**Published:** 2022-02-27

**Authors:** Kyoung Ae Kong, Young Eun Kim, Sunho Lim, Bo Young Kim, Ga Eun Kim, Soo In Kim

**Affiliations:** 1Department of Preventive Medicine, College of Medicine, Ewha Womans University, Seoul 03760, Korea; kkong@ewha.ac.kr (K.A.K.); kristyrabbit@ewhain.net (Y.E.K.); sentimenthol@ewhain.net (S.L.); 2Department of Preventive Medicine, Graduate School, Ewha Womans University, Seoul 03760, Korea; graceint222@daum.net; 3Department of Psychiatry, College of Medicine, Ewha Womans University, Seoul 03760, Korea; kgedlf@hanmail.net

**Keywords:** living alone, living with others, depressive symptoms, suicidal ideation

## Abstract

This study compared the prevalence of depressive symptoms and suicidal ideation in individuals living alone compared with those living with others and assessed the contribution of socio-demographic factors and physical health to these differences. We analyzed 2221 individuals living alone and 19,397 individuals living with others aged 20–80 years, drawn from the Korean National Health and Nutrition Examination Survey dataset in South Korea. The study group divided into three subgroups based on age to determine whether there were differences in mental health according to age. Depressive symptoms and suicidal ideation were evaluated by self-reported questionnaires. The sex- and age-adjusted prevalence rates of depressive symptoms and suicidal ideation were higher in those living alone than those living with others. The proportion of socio-economic status and physical health explaining the differences of depressive mood and suicidal ideation between the two groups was greater in the age group over 35 years old. Considering the difference in factors that explain depressive symptoms and suicidal ideation among individuals living alone in the age group over 35 years of age and younger groups under 34 years of age, policies should be developed that will address the mental health needs of each age group.

## 1. Introduction

Living alone in Korea increased from 27.2% among the total general households in 2015 to 31.7% in 2020 [1]. The number of individuals living alone is increasing every year, and this trend is occurring all over the world, especially in the 20–30-year-old and elderly groups [1,2]. The increase in young adults living alone is contributing markedly to the overall increase of individuals living alone in South Korea and has been steadily increasing since the 2000 s [3,4]. In addition, the number of elderly people has rapidly increased and many of them are living alone. Elderly individuals who live alone account for 19.63% of the total one person households in 2021, and is expected to increase to 39.2% by 2035 [5,6]. Many studies have examined the effects of living arrangement on mental health in the elderly and these studies showed that individuals living alone have a lower life satisfaction and mental health compared to those living with others [7,8,9,10]. These studies suggest that the loneliness, social isolation, poor social support and physical health and low socioeconomic status (SES) may explain these results [11]. Studies of middle-aged and young individuals living alone have shown that they do not have worse mental health than those living with others [12,13]. According to these studies [12,13], the number of middle-aged individuals who are living alone is increasing and among them, the proportion of high-educated and high income households has risen. Individuals living alone with these characteristics include ‘goose fathers’ who work alone in Korea to pay for their family’s living expenses and study abroad expenses, employees of public institutions who live separately from their spouses due to a move in a public institution and divorced people due to demographic and social changes in Korea. Additionally, the middle-aged individuals living alone are reported to have less stress and higher subjective health levels than those living with others, as individuals living with others have burdens of child education and support of elderly parents [14]. Meanwhile, employment and educational opportunities are concentrated in large cities such as Seoul, and there is a youth housing subscription system that only non-homeowners or member of those household can apply for and some younger age group become independent from their parents to increase their chances of winning the subscription system among younger age group. Therefore, the reasons for living alone are very different between the younger age group and older age groups. Although research on living alone has been steadily conducted, most studies evaluate specific diseases or physical and psychological status of specific age groups, and there have been no studies on nationally representative samples of all age groups. 

The purpose of this study was to examine whether there is a difference in mental health between individuals living alone and those living with others and to compare differences in young, middle-aged, and elderly groups. We further examined the contribution of SES or physical health to the mental health of individuals living alone in each age group.

## 2. Materials and Methods

### 2.1. Data Source and Study Subjects

This study examined the datasets of the Korea National Health and Nutrition Examination Survey (KNHANES) by the Korea Disease Control and Prevention Agency (KDCA). The KNHANES is an ongoing surveillance system on the health and nutritional status of Koreans and a nationwide cross-sectional survey conducted every year on a representative sample of the non-institutionalized population [15]. 

Through complex multi-stage stratified clustered probability sampling, 192 primary sampling units (PSU) were drawn from approximately 200,000 geographically defined units based on census track. For each PSU consisting of an average of 60 households, 20–25 households were systematically sampled from each PSU. All individuals aged ≥1 year of the selected households were included in the eligible population for the survey. A surveyor visited selected households and collected basic and contact information. The health interview and examination surveys were conducted at the KNHANES-dedicated Mobile Examination Centers across the country [16]. The number of eligible persons in each year of the survey was approximately 10,000, and the participant rates were approximately 75–78%. A sample weight was provided to consider sampling probability, non-response rate, and post-stratification to reflect the population distribution by gender and age in Korea.

In this study, we used four annual datasets from the KNHANES, which included questionnaires about depressed moods lasting 2 weeks or longer and suicidal ideation [8] over the past year: 2013, 2015, 2017, and 2019 (the first and third year of the KNHANES-Ⅵ, the second year of the KNHANES-Ⅶ, and the first year of the KNHANES-Ⅷ, respectively). The number of survey participants in the four-year dataset was 31,635 (8018, 7380, 8127, and 8110 by year, respectively) [17]. Even-year data of the KNHANES-VI, VII, and VIII were not analyzed because they were collected only from adolescents (not to adults). The KNHANES was organized into three-year units but it is designed and conducted as a rolling survey and can be analyzed by year or by integrating data from multiple years. 

Of the 24,883 subjects aged 20–80 years who participated in the health survey of 2013, 2015, 2017, or 2019, we excluded 37 subjects without information on the number of family members and marital status as well as 2095 subjects who did not respond to the questionnaires about depressed mood or SI. We further excluded 1133 subjects who did not answer the questions on educational level, occupation, income and house owning of the household, history of chronic diseases, or EQ-5D for self-rated health status. Finally, 21,618 subjects were included in the analysis.

### 2.2. Main Variables and Covariates

Living alone and living with others were defined by the number of family members and the presence of a spouse living in the same house, regardless of marital status. Individuals who had no other family member other than him/herself and who did not live with a spouse were classified as living alone. Individuals with another family member or spouse living in the same house were classified as living with others. 

As measures of mental health status, we inquired as to the experience of depressive symptoms and SI. Depressive symptoms were assessed by the question, “Have you felt sad or hopeless enough to interfere with your daily life for more than two consecutive weeks?” SI was assessed by, “Have you seriously considered suicide in the last year?” Both questions were answered in yes/no format.

Variables representing socioeconomic status and physical health status were considered covariates in the models explaining the excess prevalence of poor mental status among those living alone. SES variables were educational level (middle school graduate or below, high school graduate, or college graduate or above), house ownership (no house or ownership of ≥1 house), occupational groups (non-manual, service and sales workers, manual workers, or outside the workforce), income groups based on sex- and age-specific quintiles of the monthly household income divided by the square root of the number of family members, and residential area (Seoul and Gyeonggi-do, other cities, and rural area). 

As physical health status, we used the number of chronic illnesses and the four dimensions of the EQ-5D. The number of chronic illness was defined as the number of diseases that were diagnosed by a physician and the diseases included cancer, stroke, myocardial infarction or angina, osteoarthritis or rheumatic arthritis, thyroid diseases, diabetes, hypertension, asthma, renal failure, and hepatitis B or hepatitis C or liver cirrhosis. Other variables for physical health status were assessed by questions about mobility, self-care, usual activities, and pain/discomfort of the EQ-5D with a 3-level-response (no problems, some problems, and extreme problems). EQ-5D was developed as a measure that generates a single index value for health status by EuroQoL Group. We excluded the dimension of anxiety/depression and used the other four dimensions separately because mental health status was the main outcome of this study.

In the analysis of individuals who were 20–34 years old, the elementary and middle school education categories were integrated and the EQ-5D health status items were used in combination with the two levels of some or extreme problem

### 2.3. Statistical Analysis 

Data are presented as frequency and the proportion of the sociodemographic characteristics of those living alone and those living with others. The proportion of the depressive symptoms and the SI according to the categories of each sociodemographic factor are presented and were compared through prevalence ratios (PR) by the generalized linear model with binomial distribution and log link. Sex- and age-adjusted prevalence of depressive symptoms and SI of those living alone and those living with others were estimated based on the even distribution of sex and all 5-year age groups (using the SAS SURVEY REG procedure). In accordance with the complex sampling design, the proportion of the variables and the prevalence of mental status were estimated by incorporating the sampling weight. Variances of all estimates were calculated using the Taylor linearization method.

We assessed the association between the type of living (living alone/living with others) and the mental status using a generalized linear model with binomial distribution and log link by three age groups of 20–34, 35–64, and 65–80 years old [18]. The base model included the type of living (living alone/living with others) and sex and 5-year age groups and the full model additionally included all variables of SES and physical health. The contribution of the SES and physical illness on the excess prevalence of depressive symptoms and SI were assessed by explained fractions (EFs, %), i.e., the proportion of excess prevalence explained by each variable or the combination of the variables. EFs were calculated as [(PR-1) − (PRa-1)]/(PR-1), where PR is the that of those living alone from the base model and PRa is the PR of those living alone in the model including the potential mediating or confounding variables in addition to the variables in the base model. The contribution of the SES and physical illness were assessed for the full model as well as for the models including all SES variables, each SES variable, and variables representing physical health status. All statistical analyses were performed with SAS software (version 9.2, SAS Institute, Cary, NC, USA). 

## 3. Results

### 3.1. Sociodemographic Characteristics of Participants 

Table 1 shows the socio-demographic characteristics of those living alone and those living with others. 

The living alone group was older than the living with others group, and these individuals were more likely to belong to the lower socioeconomic group in all three age groups. A higher proportion of the living alone group was less educated, belonged to the lowest income quintile, did not own their family’s house, and was outside the workforce. The living alone group had more chronic diseases and a higher proportion of the poor health status than that of those living with others. Some characteristics, including sex, differed by age groups. In the 20–34 and 35–64 years groups, there were more men among those living alone than among those living with others, but in the group 65–80 years of age, 82% of those living alone were women compared with approximately 50% of those living with others. In the 20–34 years, unlike the other two age groups, the proportions of the subjects graduating (or attending) college, those with a non-manual occupation, and metropolitan (Seoul/Gyeonggi area) residents were higher and the proportion of unemployed was lower in the living alone group than in the living with others group. 

### 3.2. Association between Sociodemographic Factors and Mental Health 

Table 2 shows the association between each socio-demographic factor and mental health.

Women had a higher prevalence of depressive symptoms than men (OR 1.61, 95% CI 1.49–1.74). Lower education, lower income, not owning a house, service and sales, manual occupation, or having no job (compared with non-manual occupation) were associated with depressive symptoms, with PRs ranging from 2.11 (no job relative to non-manual occupation) to 2.27 (elementary relative to college graduate). More frequent chronic diseases and poorer health status with problems in mobility, self-care, usual activities, and pain/discomfort were also associated with depressive symptoms, with PRs ranging from 2.90 (≥4 diseases relative to no disease) to 4.14 (extreme relative to no problem in mobility). All these factors except sex were more strongly associated with SI than depressive symptoms, with ORs of 3.11 to 8.08.

The sex- and age-adjusted prevalence rates of depressive symptoms were 11.1% in those living with others and 17.6% in those living alone (Table 3). 

In the living with others group, the prevalence of depressive symptoms was the highest in the 65–80 years group (12.9%), followed by the 20–34 years group (11.8%) and the 35–64 years group (9.9%). In contrast, among those living alone, an older age group tended to have a higher prevalence of depressive symptoms, and the prevalence of the 35–64 years group was very similar to that of the 65–80 years group (20.5%). The sex and age-adjusted prevalence of the SI was 4.6% in those living with others and 9.0% in those living alone. The prevalence pattern by age group was similar to that of depressive symptoms and the prevalence among those living alone was higher than those living with others.

### 3.3. Association between Those Living alone and Sociodemographic Factors and Mental Health by Age Group

In the sex and age-adjusted model (base model) and in the model including all SES and health status variables, the association between mental health and the living arrangement and other variables are presented in Table 4. 

### 3.4. Explained Proportion by the Socioeconomic Factors and Physical Health 

Table 5 shows the depressive symptoms and SI of those living alone relative to those living with others and the proportion by SES and physical health by age group. 

The PR of the depressive symptoms of those living alone to those living with others was the largest among those aged 35–64 years (PR 1.97, 95% CI 1.69–2.29). When the SES of educational level, occupation, household income, house owning, and residential area were adjusted, the PR was 1.39, which explained 60% of the excess prevalence because of living alone. Physical health, including the number of chronic diseases, problems with mobility, self-care, usual activities, and pain/discomfort, explained 38% of the excess prevalence, and both SES and physical health accounted for 73% of the excess prevalence by living alone (PR 1.27, 95% CI 1.08–1.49). For the 65–80 years group, the PR of depressive symptoms in those living alone to those living with others was 1.52 (95% CI 1.31–1.75). SES explained 65% of the excess prevalence, which was similar to that observed in 35–64 years group; physical health explained 14% and SES and physical health as a whole explained 66%. The degree of explanation according to education level (15% in 35–64 years group vs 7% in 65–80 years group) and physical health (38% in 35–64 years group vs 14% in 65–80 years group) was lower than that of the 35–64 years old group. The prevalence of depressive symptoms in those living alone in the 20–34 years group was 1.22 (95% CI 0.92–1.62), which was not statistically significant; this relationship could not be explained when variables corresponding to SES were included in this model. In the model including education level or occupation, the PR of the depressive symptoms in those living alone for those living with others was approximately 1.30, which was higher than in the basic model, indicating that these variables act differently from those of the age groups of 35–64 or 65–80 years. 

For suicidal ideation, the PR of those living alone to living with others in the 35–64 years group was 3.01 (95% CI 2.44–3.73). When all SES variables were adjusted, the PR was 1.77, which explained 62% of the excess prevalence from living alone. Physical health explained 42% of the excess prevalence, and when both SES and physical health were included, the PR was 1.55 (95% CI 1.22–1.96) and 73% of the excess prevalence by living alone was explained. In the 65–80 years group, the PR of the depressive symptoms of those living alone to those living with others was 1.53 (95% CI 1.21–1.93). SES including educational level, occupation, household income, house owning and residential area explained 98% of the excess prevalence from living alone. Among the SES variables, the explained proportions of household income and house-owning were particularly large at 67% and 52%, respectively. Physical health explained 25%, and both SES and physical health explained 100% of the excess prevalence of SI of living alone. The prevalence of SI in those living alone compared to that of those living with others in the 20–34 years group was 1.68 times (95% CI 1.10–2.57). However, physical health only explained 10% of the excess prevalence of SI of living alone, and SES did not explain it (in the model including 5 SES variables, PR 1.81, 95% CI 1.15–2.85). When each variable of SES was included in the model, household income explained 28% and house owning explained 40% of the excess prevalence of SI. The PR of SI in living alone relative to living with others increased to 1.87 and 1.85 when educational level and occupation were adjusted, respectively.

## 4. Discussion

This large population-based study using nationally representative datasets of KNHANES showed that depressive symptoms and SI were more prevalent among those living alone than in those living with others among all age groups. The prevalence of depressive symptoms and SI in living alone is similarly high among individuals in the 65–80 and 35–64 years groups and the difference with those living with others is greatest in those aged 35–64 years. We estimated the contribution of factors, including age-specific SES and physical health, to the excess risk of depressive symptoms and SI in individuals living alone. SES explained the differences in depressive symptoms the most between those living alone and those living with others in the 65–80 and 35–64 years groups, and the proportions explained by physical health was relatively high in the 35–64 years group compared with that in other age groups. However, the 20–34 years group showed a different pattern from other age groups, in which the difference between the two groups was not well explained by SES and physical health. Additionally, the difference in the prevalence of depressive symptoms and SI between individuals living alone and those living with others was smallest in this group. This is probably explained by the difference in SES from other age groups. 

The socioeconomic positions of 20–34-year-olds living alone differed according to SES, whereas individuals living alone in other age groups had poor SES across all indicators. The cause of the difference in SES between the 20–34-year-old group and the 35–80-year-old group is that most of the younger age group had graduated from college, worked in non-manual jobs, and lived in metropolitan area due to Korea’s socioeconomic growth and high enthusiasm for education. The high proportions of college graduates and non-manual job workers among individuals living alone and aged 20–34 years might be the reason why those living alone do not show negative mental health outcomes. Therefore, it is thought that factors such as education level, occupation, and residential area did not significantly contribute to the difference in mental health between individuals of the younger generation living alone and those living with others. However, house income and house ownership partially explained the differences in mental health between these two groups. In Korea, it has long been a custom for unmarried adult children to live with their parents and remain a member of their parents’ household. Therefore, the characteristics of individuals 20–34 years of age living alone, many of whom are college graduate professionals who live in a metropolitan area might reflect situations or conditions in which they had to form their own household independent of their parents. There are also incentives such as priority of housing subscription given when a household is separated from its parents. Additionally, young individuals living alone are more likely to maintain social relationships with family and friends than are middle-aged or elderly living alone. Thus, social isolation is expected to differ by age3.

The number of young individuals living alone is continuously increasing. This trend is the same in the UK, Europe, and Japan. Late marriage is cited as the reason for the increase in young individuals living alone. The reason for this view is that economic conditions must be met to get married and form a family, and as the unemployment rate rises due to job instability and economic stagnation, marriage in this age group is delayed [4]. We also found that individuals 20–34 years of age living alone have lower SES in other aspects, such as income quintile or house ownership. These individuals had to living alone, but this does not mean that they are in a relaxed economic situation. These individuals are probably the only source of income for their household and are responsible for their housing and living expenses. Emotional and financial help and interaction from parents and other family members would have decreased. Currently, employment in Korea is unstable, and the low employment rate, high unemployment rate, and increase in non-regular workers are very big issues; it is difficult to maintain such a stable job and income. Since these individuals are the only source of income in the household, the loss of a job will represent an immediate crisis [20]. The high low SES rate in this aspect and the absence of a companion/family to help take care of the household responsibilities may explain the poor mental health of individuals living alone. As such, in the 20–34 years of age group, various SES factors intertwined, making the difference between those living alone and those living with others inconspicuous, and it seems that the degree and direction of explanation vary according to each SES.

Low SES is a well-known risk factor for negative mental health irrespective of age and sex [21,22] and has been identified in SES indicators such as income, house ownership, and education level. Mental health problems of those living alone may be a problem of the mental health gap according to SES, and according to this study, more than three-fifths of the mental health difference between those living alone and those living with someone over the age of 35 is explained by SES. Among SES, income and house ownership accounted for a large part of negative mental health, which shows that social support policies for housing and income stability are the most needed in Korea, and this has been confirmed in other studies [23,24,25]. Physical health explained the difference between the two groups at all ages, especially in the 35–64 years group. Since 35–64 years are the most economically and socially active years, depressive symptoms or SI may have been greater if physical health was not good. However, poor physical condition may be the reason for low SES or living alone. Although Korea has a well-equipped universal health insurance system, the social support for acute diseases, accidents, and irreversible physical disabilities is relatively weak compared with that for chronic severe disabilities such as dementia or stroke. Therefore, it is necessary to strengthen the social support system for physical problems that are relatively lacking in support.

Studies have shown that living alone is not a risk factor for mental health, and social isolation or loneliness is the main problem [12,26]. Social isolation is defined as having few contacts, little involvement in social activities and living alone [27]. Loneliness refers to a subjective feeling of dissatisfaction with one’s social relationships [27]. Social isolation and loneliness have been linked with increased mortality [28], depression, cardiovascular disease and poor quality of life and wellbeing [29]. Michale et al. showed that the mental health of women living alone was better than that of women living with others and the number of social network connections, and the level of social participation of women living alone was similar to those of women living with a spouse. The authors suggested that social isolation, rather than living alone, has a negative effect on mental health [12]. De Vaus et al. also reported that living alone have some negative effects on loneliness and life satisfaction; however, these effects are generally modest. These authors suggested that the differences could not be attributed living alone, rather that they were from different characteristics such as social connection of individuals living alone [26]. We were unable to examine the effect of social isolation or loneliness because social isolation or loneliness is not investigated in the KNHANES. However, poor SES and physical health could reduce the number of social participations or social network connections and cause social isolation. In this study, differences in mental health between those living alone and those living with others were not fully explained by SES and physical health. Therefore, the poor mental health of individuals living alone in this study may be influenced by SES and physical health as well as related social isolation and loneliness. Since social isolation and network are very important components of mental health, these could explain the differences in mental health between those living alone and those living with others. Our findings, in which most of the negative mental health of individuals aged 35 years and older individuals living alone was explained by SES, indirectly reflect the contribution of social isolation and loneliness. Our results are probably because middle-aged and older generations who are accustomed to Korea’s long-standing extended family customs experience a greater psychological and economic burden in response to living alone, and the social support system for the middle-aged and elderly living alone in Korea is poorer than other countries with strong support systems. Middle-aged and older individuals living alone are more likely to experience emotional confusion and depression from negative life events such as divorce, separation, and bereavement, and this situation is closely related to low SES. In addition, substance dependence problems such as smoking and drinking are also high, which is also associated with low SES, and aggravates negative mental health problems [13].

In addition, the extent to which physical health explains the difference in mental health between individuals living alone and those living with others was greater in the middle-aged group than in the elderly group. This difference may be because the elderly has various physical health problems among both individuals living alone and those living with others. Another explanation may be that physical health is regarded as a more important issue among middle-aged individuals than that in the elderly. Henning-Smith et al. reported that middle-aged individuals (ages 35–64) living alone showed significantly worse health than those living with others [30], which is consistent with the results of our study. In addition, the physical health problems of the middle-aged group may be related to low SES or lack of social support and may have been the cause of living alone. Lee et al. showed that middle-aged individuals living alone are in a difficult socio-economic environment and in poor health conditions and residential environments as much as the elderly living alone [31]. Therefore, it will be necessary to provide services to protect the physical and mental health in middle aged and elderly individuals living alone and establish a social network for those living alone who have difficulties in forming social relationships. This study has several limitations. First, since the design of this study was cross-sectional, we could not form conclusions with regard to causality. Second, it was based on self-reported questionnaires that impose real limits on the objective evaluation of mental health. Third, our findings were not reliable as longitudinal data as they were based on one-time evaluations. Additionally, recall biases and denials must be considered. Fourth, depressive symptoms and suicidal ideation were evaluated by very simplified questions. The limitations of this measure are as follows. There is a possibility that participants who were assessed as depressed might not have been clinically depressed. Finally, social isolation or social networks have not been investigated, which are thought to be able to explain mental health in living alone to some extent, so it is not possible to confirm how these are intertwined with SES and physical health and affect mental health. Despite these limitations, our study had several strengths. A single question about depressive mood over a relative short period (two weeks) might decrease the possibility of reverse causality from having depression to being living alone than using the diagnosis of depression based on DSM 5. Wittchen et al. reported that use of a single question to screen for major depression had a modest impact on the recognition of depression in the general population [31]. Several studies have reported that a single question about depressive mood is very meaningful and world-wide screening questions for detecting those symptoms [32,33,34]. Additionally, a single screening questions is brief and reduces the time needed for a nationwide survey [31]. This study used a large and nationally representative sample of both individuals living alone and those living with others. Furthermore, to the best of our knowledge, this is the first study in Asia to report a difference in the contributions of SES and physical health to mental health between individuals living alone and those living with others by age group.

## 5. Conclusions

This study revealed that individuals living alone had more depressive symptoms and SI than those living with others. SES best explained the differences in depressive symptoms between those living alone and those living with others in the 35–80-year group, while the proportion explained by physical health was relatively high in the 35–64-year group. However, the 20–34-year group showed a different pattern from other age groups, and the difference between the two groups was not well explained by SES or physical health. Therefore, it is necessary to identify the needs of each age group and provide services according to their needs to improve the mental health of individuals living alone.

## Figures and Tables

**Table 1 diagnostics-12-00603-t001:** Sociodemographic characteristics of individuals living alone and those living with others in South Korea.

	Age 20–34 Years		Age 35–64 Years		Age 65–80 Years	
	Living with Others	Living Alone	Living with Others	Living Alone	Living with Others	Living Alone
	(*n* = 3580)	(*n* = 349)	(*n* = 11530)	(*n* = 721)	(*n* = 4287)	(*n* = 1151)
Characteristics	*n* (%)	*n* (%)	*n* (%)	*n* (%)	*n* (%)	*n* (%)
Age (years)						
20–24	34.1 (1.0)	28.6 (3.5)				
25–29	31.8 (0.9)	38.8 (3.3)				
30–34	34.1 (1.1)	32.6 (3.6)				
35–39			16.7 (0.5)	16.4 (1.8)		
40–44			17.7 (0.5)	12.3 (1.6)		
45–49			18.8 (0.5)	13.8 (1.5)		
50–54			17.9 (0.4)	16.1 (1.7)		
55–59			16.8 (0.5)	19.1 (1.6)		
60–64			12.2 (0.4)	22.3 (1.7)		
65–69					36.5 (0.9)	20.7 (1.2)
70–74					27.7 (0.8)	23.0 (1.3)
75–80					35.8 (1.0)	56.3 (1.7)
Sex						
Men	50.6 (0.9)	68.7 (2.7)	48.9 (0.4)	58.7 (2.1)	48.5 (0.8)	18.1 (1.3)
Women	49.4 (0.9)	31.3 (2.7)	51.1 (0.4)	41.3 (2.1)	51.5 (0.8)	81.9 (1.3)
Education						
≥College	77.2 (0.9)	83.4 (2.4)	42.9 (0.9)	30.9 (2.2)	10.2 (0.7)	3.9 (0.6)
High	20.9 (0.8)	15.8 (2.3)	37.8 (0.7)	34.7 (2.1)	18.5 (0.7)	10.5 (0.9)
Middle	1.4 (0.2)	0.8 (0.5)	10.2 (0.4)	16.4 (1.5)	15.0 (0.6)	10.7 (1.0)
≤Elementary	0.4 (0.1)	0.0 (0.0)	9.1 (0.4)	18.0 (1.6)	56.4 (1.1)	74.9 (1.4)
Income quintile						
Q5 (high)	21.1 (1.0)	6.7 (1.4)	20.6 (0.7)	11.8 (1.6)	24.8 (1.0)	5.2 (0.7)
Q4	20.6 (0.9)	14.7 (2.2)	21.3 (0.6)	10.7 (1.4)	22.4 (0.8)	13.3 (1.1)
Q3	20.6 (0.9)	24.2 (2.9)	20.3 (0.5)	13.2 (1.5)	18.2 (0.8)	22.4 (1.4)
Q2	20.4 (0.9)	19.8 (2.5)	19.8 (0.6)	20.3 (1.9)	17.1 (0.8)	30.2 (1.5)
Q1 (low)	17.4 (0.8)	34.5 (3.4)	18.0 (0.6)	44.0 (2.3)	17.5 (0.8)	28.9 (1.5)
House owning						
Yes	60.3 (1.3)	5.5 (1.3)	70.8 (0.9)	29.6 (2.0)	81.4 (0.9)	44.6 (2.0)
No	39.7 (1.3)	94.5 (1.3)	29.2 (0.9)	70.4 (2.0)	18.6 (0.9)	55.4 (2.0)
Occupation						
Non-manual	36.9 (0.9)	52.0 (3.5)	30.6 (0.7)	23.1 (2.0)	3.7 (0.3)	0.6 (0.3)
Service and sales	13.7 (0.7)	12.5 (1.7)	15.3 (0.4)	12.6 (1.5)	4.6 (0.4)	3.6 (0.6)
Manual	11.7 (0.6)	11.9 (2.0)	27.7 (0.6)	31.9 (2.1)	25.0 (1.0)	21.8 (1.4)
Outside the workforce	37.8 (1.0)	23.5 (3.2)	26.4 (0.5)	32.3 (2.2)	66.7 (1.0)	74.0 (1.5)
Residential area						
Metropolitan	47.3 (1.2)	57.4 (5.0)	45.8 (0.8)	43.9 (2.7)	40.3 (1.3)	33.4 (1.9)
City	28.5 (1.3)	18.2 (3.8)	26.5 (1.1)	24.6 (2.4)	24.9 (1.3)	21.6 (1.9)
Rural	24.2 (1.4)	24.4 (4.4)	27.8 (1.2)	31.5 (2.7)	34.8 (1.5)	45.0 (2.3)
Number of chronic diseases						
0	95.6 (0.4)	93 (1.8)	71.9 (0.5)	61.6 (2.2)	26.0 (0.8)	23.0 (1.4)
1	4.1 (0.4)	6.4 (1.8)	20.4 (0.4)	25.1 (1.9)	36.9 (0.8)	32.1 (1.6)
2	0.3 (0.1)	0.6 (0.4)	6.1 (0.3)	9.1 (1.2)	25.2 (0.8)	29.4 (1.5)
3			1.3 (0.1)	3.6 (0.8)	9.8 (0.5)	12.8 (1.1)
4~6			0.3 (0.1)	0.6 (0.2)	2.2 (0.3)	2.8 (0.5)
Mobility						
No problem	97.4 (0.3)	96.5 (1.2)	92.7 (0.3)	85.4 (1.5)	65.0 (0.9)	50.8 (1.7)
Some problems	2.6 (0.3)	3.6 (1.2)	7.2 (0.3)	13.6 (1.5)	33.6 (0.9)	44.8 (1.7)
Extreme			0.2 (0.0)	0.9 (0.4)	1.4 (0.2)	4.4 (0.6)
problems						
Self-care						
No problem	99.6 (0.1)	98.5 (0.8)	98.2 (0.1)	95.1 (0.9)	90.3 (0.5)	82.3 (1.3)
Some problems	0.4 (0.1)	1.5 (0.8)	1.7 (0.1)	4.7 (0.9)	8.5 (0.5)	16.4 (1.3)
Extreme problems			0.1 (0.0)	0.2 (0.1)	1.2 (0.2)	1.3 (0.4)
Usual activities						
No problem	97.3 (0.3)	98.0 (0.9)	95.6 (0.2)	86.8 (1.4)	80.7 (0.7)	71.3 (1.6)
Some problems	2.7 (0.3)	2.0 (0.9)	4.3 (0.2)	12.8 (1.4)	17.6 (0.7)	26.5 (1.5)
Extreme			0.1 (0.0)	0.4 (0.2)	1.7 (0.2)	2.2 (0.5)
problems						
Pain/discomfort						
No problem	85.7 (0.7)	83.6 (2.1)	80.3 (0.4)	71.1 (2.0)	64.3 (0.9)	53.5 (1.7)
Some problems	14.3 (0.7)	16.4 (2.1)	18.9 (0.4)	25.2 (1.8)	30.2 (0.8)	36.8 (1.6)
Extreme			0.9 (0.1)	3.7 (0.8)	5.4 (0.4)	9.7 (1.0)
problems						

Metropolitan: Seoul and Gyonggi-do.

**Table 2 diagnostics-12-00603-t002:** Association between sociodemographic factors and depressive symptoms and suicidal ideation.

			Depressive Symptoms			Suicidal Ideation		
		*n*	%	PR	(95% CI)	%	PR	(95% CI)
Age (years)	20–24	1261	11.8	1.00	(reference)	3.5	1.00	(reference)
	25–29	1221	14.0	1.10	(0.90–1.35)	5.7	1.31	(0.92–1.86)
	30–34	1447	9.6	0.86	(0.70–1.06)	3.5	0.94	(0.65–1.35)
	35–39	1891	7.3	0.62	(0.50–0.76)	2.1	0.58	(0.39–0.85)
	40–44	2051	8.3	0.64	(0.52–0.79)	3.0	0.71	(0.49–1.02)
	45–49	1980	9.8	0.75	(0.61–0.91)	3.7	0.83	(0.58–1.18)
	50–54	2091	10.8	0.91	(0.75–1.10)	4.7	1.05	(0.75–1.46)
	55–59	2209	12.8	1.05	(0.87–1.26)	5.6	1.42	(1.04–1.94)
	60–64	2029	13.4	1.12	(0.93–1.34)	5.9	1.48	(1.08–2.02)
	65–69	1817	14.3	1.20	(1.00–1.44)	6.2	1.62	(1.19–2.22)
	70–74	1573	14.1	1.21	(1.00–1.46)	7.9	1.71	(1.24–2.35)
	75–80	2048	15.7	1.29	(1.08–1.54)	8.1	1.87	(1.38–2.53)
Sex	Men	9365	8.4	1.00	(reference)	3.9	1.00	(reference)
	Women	12253	14.2	1.61	(1.49–1.74)	5.4	1.28	(1.13–1.44)
Education	≥College	8501	8.7	1.00	(reference)	2.4	1.00	(reference)
	High	6224	11.1	1.34	(1.21–1.48)	5.0	1.98	(1.67–2.35)
	Middle	2212	13.4	1.59	(1.40–1.80)	6.7	2.60	(2.12–3.19)
	≤Elementary	4681	18.9	2.27	(2.07–2.48)	9.5	3.58	(3.05–4.20)
Income quintile	Q5 (high)	4384	8.8	1.00	(reference)	2.6	1.00	(reference)
	Q4	4391	9.1	1.07	(0.93–1.22)	3.1	1.25	(0.98–1.59)
	Q3	4344	10.5	1.27	(1.11–1.44)	3.7	1.60	(1.27–2.01)
	Q2	4291	11.5	1.51	(1.34–1.71)	5.8	2.42	(1.95–2.99)
	Q1 (low)	4208	17.3	2.12	(1.89–2.38)	8.4	3.52	(2.87–4.31)
House owning	Yes	14648	10.4	1.00	(reference)	3.8	1.00	(reference)
	No	6970	13.4	1.36	(1.26–1.46)	6.3	1.72	(1.54–1.93)
Occupation	Non-manual	5231	7.2	1.00	(reference)	2.3	1.00	(reference)
	Service and sales	2651	11.8	1.67	(1.45–1.93)	3.9	1.89	(1.47–2.43)
	Manual	5078	10.9	1.56	(1.38–1.76)	4.9	2.21	(1.78–2.73)
	Outside the workforce	8658	14.8	2.11	(1.89–2.35)	6.7	3.11	(2.57–3.78)
Residential area	Metropolitan	9573	10.5	1.00	(reference)	4.4	1.00	(reference)
	City	5632	11.9	1.18	(1.08–1.30)	4.6	1.12	(0.97–1.30)
	Rural	6413	12.5	1.34	(1.23–1.46)	5.2	1.33	(1.16–1.52)
Number of	0	13625	10.0	1.00	(reference)	3.6	1.00	(reference)
chronic diseases	1	4797	13.0	1.30	(1.19–1.42)	6.2	1.62	(1.41–1.86)
	2	2274	16.6	1.66	(1.50–1.85)	7.6	1.87	(1.58–2.22)
	3	756	17.7	1.92	(1.65–2.24)	9.6	2.59	(2.06–3.26)
	4~	166	29.1	2.90	(2.28–3.69)	17.5	4.82	(3.47–6.69)
Mobility	No problem	18408	9.7	1.00	(reference)	3.7	1.00	(reference)
	Some problems	3063	24.1	2.41	(2.23–2.61)	12.0	3.10	(2.74–3.50)
	Extreme problems	147	35.6	4.14	(3.39–5.06)	27.8	8.08	(6.27–10.42)
Self-care	No problem	20705	10.9	1.00	(reference)	4.3	1.00	(reference)
	Some problems	836	28.8	2.63	(2.36–2.94)	17.4	3.76	(3.20–4.42)
	Extreme problems	77	27.7	2.54	(1.78–3.62)	23.3	4.86	(3.18–7.42)
Usual activities	No problem	19725	10.1	1.00	(reference)	3.8	1.00	(reference)
	Some problems	1774	28.2	2.67	(2.45–2.91)	15.6	3.89	(3.42–4.42)
	Extreme problems	119	40.3	3.78	(3.02–4.74)	30.4	7.56	(5.71–10.02)
Pain/discomfort	No problem	16344	9.0	1.00	(reference)	3.2	1.00	(reference)
	Some problems	4792	18.9	2.08	(1.93–2.25)	8.8	2.65	(2.35–2.99)
	Extreme problems	482	33.6	3.70	(3.24–4.23)	23.6	6.62	(5.51–7.96)

Metropolitan: Seoul and Gyonggi-do. PR, prevalence ratio; CI, confidence intervals.

**Table 3 diagnostics-12-00603-t003:** Prevalence of depressive symptoms and suicidal ideation among those living alone and those living with others.

	Living with Others		Living Alone	
	N	Prevalence (SE) [19]	*n*	Prevalence (SE) [19]
**Depressive Symptoms**				
All ages	19,397	11.1 (0.3)	2221	17.6 (1.0)
20–34 years	3732	11.8 (0.6)	366	13.2 (1.7)
35–64 years	12,099	9.9 (0.3)	757	19.3 (1.6)
65–80 years	4587	12.9 (0.6)	1234	20.5 (1.5)
**Suicide Ideation**				
All ages	19,397	4.6 (0.2)	2221	9.0 (0.7)
20–34 years	3732	4.1 (0.4)	366	6.0 (1.3)
35–64 years	12,099	3.8 (0.2)	757	11.2 (1.3)
65–80 years	4587	6.8 (0.5)	1234	10.0 (1.1)

SE, standard error. Estimated prevalence based on the distribution of the sex and 5-year age group of the population summed living alone and living together.

**Table 4 diagnostics-12-00603-t004:** Association between those living alone and sociodemographic factors and mental health by age group.

		Depressive Symptoms						Suicidal Ideation					
		20–34 Years		35–64 Years		65–80 Years		20–34 Years		35–64 Years		65–80 Years	
		PR	(95% CI)	PR	(95% CI)	PR	(95% CI)	PR	(95% CI)	PR	(95% CI)	PR	(95% CI)
**Model including living alone, sex, and 5-year age group**													
Family type	Living with others	1.00	(reference)	1.00	(reference)	1.00	(reference)	1.00	(reference)	1.00	(reference)	1.00	(reference)
	Living alone	1.22	(0.92–1.62)	1.97	(1.69–2.29)	1.52	(1.31–1.75)	1.68	(1.10–2.57)	3.01	(2.44–3.73)	1.53	(1.21–1.93)
**Model including living alone, all listed variables below, sex, and 5-year age group**													
Family type	Living with others	1.00	(reference)	1.00	(reference)	1.00	(reference)	1.00	(reference)	1.00	(reference)	1.00	(reference)
	Living alone	1.37	(1.01–1.85)	1.27	(1.08–1.49)	1.17	(1.01–1.36)	1.82	(1.16–2.84)	1.55	(1.22–1.96)	1.00	(0.80–1.25)
Education	≥College	1.00	(reference)	1.00	(reference)	1.00	(reference)	1.00	(reference)	1.00	(reference)	1.00	(reference)
	High school	1.29	(1.06–1.58)	1.28	(1.10–1.48)	1.01	(0.71–1.43)	2.14	(1.54–2.96)	1.64	(1.25–2.14)	1.35	(0.79–2.28)
	Middle school	1.69	(1.08–2.65)	1.33	(1.09–1.63)	1.01	(0.70–1.44)	4.73	(2.71–8.24)	1.81	(1.29–2.55)	1.20	(0.70–2.07)
	≤Elementary school			1.51	(1.24–1.84)	1.20	(0.86–1.66)			2.11	(1.50–2.97)	1.38	(0.83–2.30)
Household income	Q5(high)	1.00	(reference)	1.00	(reference)	1.00	(reference)	1.00	(reference)	1.00	(reference)	1.00	(reference)
quintiles	Q4	0.95	(0.72–1.25)	0.90	(0.74–1.08)	1.17	(0.92–1.51)	0.80	(0.47–1.37)	1.21	(0.84–1.73)	1.20	(0.79–1.83)
	Q3	0.95	(0.73–1.25)	0.97	(0.81–1.17)	1.32	(1.04–1.69)	0.72	(0.43–1.21)	1.16	(0.81–1.66)	1.81	(1.23–2.67)
	Q2	0.85	(0.64–1.13)	1.12	(0.93–1.34)	1.57	(1.24–1.99)	1.39	(0.86–2.24)	1.60	(1.14–2.24)	2.03	(1.39–2.98)
	Q1(low)	0.97	(0.73–1.29)	1.50	(1.25–1.79)	1.69	(1.34–2.14)	1.04	(0.63–1.70)	1.97	(1.41–2.76)	2.48	(1.70–3.62)
House owning	Yes	1.00	(reference)	1.00	(reference)	1.00	(reference)	1.00	(reference)	1.00	(reference)	1.00	(reference)
	No	0.92	(0.77–1.11)	1.15	(1.03–1.29)	1.20	(1.05–1.37)	1.09	(0.79–1.50)	1.30	(1.09–1.56)	1.39	(1.14–1.69)
Occupation	Non-manual	1.00	(reference)	1.00	(reference)	1.00	(reference)	1.00	(reference)	1.00	(reference)	1.00	(reference)
	Service and sales	1.56	(1.19–2.05)	1.11	(0.91–1.34)	2.15	(0.92–5.00)	1.28	(0.81–2.03)	1.24	(0.87–1.78)	0.88	(0.31–2.48)
	Manual	1.50	(1.09–2.08)	0.99	(0.83–1.19)	2.10	(0.94–4.68)	1.06	(0.61–1.84)	1.27	(0.91–1.76)	1.06	(0.43–2.60)
	Outside the workforce	1.61	(1.30–2.00)	1.12	(0.95–1.33)	2.26	(1.02–5.01)	1.49	(1.03–2.15)	1.47	(1.07–2.02)	1.21	(0.50–2.92)
Residential area	Metropolitan	1.00	(reference)	1.00	(reference)	1.00	(reference)	1.00	(reference)	1.00	(reference)	1.00	(reference)
	City	1.03	(0.85–1.25)	1.06	(0.93–1.20)	1.43	(1.21–1.69)	1.29	(0.93–1.77)	1.07	(0.87–1.31)	0.92	(0.71–1.19)
	Rural	0.86	(0.69–1.07)	1.10	(0.98–1.24)	1.48	(1.27–1.73)	0.90	(0.61–1.32)	1.03	(0.85–1.26)	1.09	(0.87–1.36)
Number of	0	1.00	(reference)	1.00	(reference)	1.00	(reference)	1.00	(reference)	1.00	(reference)	1.00	(reference)
chronic diseases	1	1.44	(1.06–1.98)	1.04	(0.91–1.18)	1.00	(0.84–1.19)	1.71	(1.06–2.76)	1.13	(0.92–1.39)	1.01	(0.78–1.32)
	2	1.63	(0.72–3.67)	1.11	(0.93–1.33)	1.04	(0.87–1.24)	1.35	(0.24–7.75)	0.98	(0.74–1.31)	0.93	(0.70–1.23)
	3			0.92	(0.68–1.25)	1.03	(0.82–1.30)			0.95	(0.62–1.45)	0.99	(0.71–1.38)
	4~			1.19	(0.73–1.94)	1.40	(1.01–1.93)			1.38	(0.70–2.75)	1.71	(1.13–2.60)
Mobility	No problem	1.00	(reference)	1.00	(reference)	1.00	(reference)	1.00	(reference)	1.00	(reference)	1.00	(reference)
	Some problems	1.72	(1.22–2.41)	1.42	(1.19–1.70)	1.23	(1.04–1.45)	1.30	(0.72–2.34)	1.32	(1.00–1.75)	1.21	(0.93–1.58)
	Extreme problems			1.70	(1.13–2.58)	1.45	(1.06–1.99)			1.66	(0.90–3.07)	1.83	(1.19–2.82)
Self-care	No problem	1.00	(reference)	1.00	(reference)	1.00	(reference)	1.00	(reference)	1.00	(reference)	1.00	(reference)
	Some problems	0.91	(0.47–1.76)	1.08	(0.86–1.35)	1.20	(1.00–1.44)	1.35	(0.55–3.29)	0.92	(0.66–1.28)	1.35	(1.03–1.75)
	Extreme problems			1.23	(0.60–2.52)	0.84	(0.51–1.40)			0.33	(0.13–0.84)	1.47	(0.86–2.52)
Usual activities	No problem	1.00	(reference)	1.00	(reference)	1.00	(reference)	1.00	(reference)	1.00	(reference)	1.00	(reference)
	Some problems	1.56	(1.07–2.29)	1.39	(1.14–1.70)	1.14	(0.95–1.35)	2.70	(1.44–5.07)	1.84	(1.34–2.52)	1.14	(0.87–1.49)
	Extreme problems			1.98	(1.26–3.11)	1.25	(0.86–1.82)			3.13	(1.64–5.99)	1.47	(0.93–2.32)
Pain/discomfort	No problem	1.00	(reference)	1.00	(reference)	1.00	(reference)	1.00	(reference)	1.00	(reference)	1.00	(reference)
	Some problems	1.62	(1.31–1.99)	1.40	(1.23–1.60)	1.43	(1.22–1.68)	1.76	(1.23–2.52)	1.62	(1.31–2.00)	1.66	(1.29–2.13)
	Extreme problems			1.64	(1.24–2.17)	1.70	(1.35–2.13)			2.21	(1.51–3.22)	2.49	(1.80–3.46)
Sex	Men	1.00	(reference)	1.00	(reference)	1.00	(reference)	1.00	(reference)	1.00	(reference)	1.00	(reference)
	Women	1.72	(1.43–2.07)	1.39	(1.24–1.56)	1.27	(1.08–1.48)	1.71	(1.24–2.35)	1.05	(0.88–1.26)	0.85	(0.68–1.07)

**Table 5 diagnostics-12-00603-t005:** Explained proportion of the socioeconomic factors and physical health in the excess prevalence of depressive symptoms or suicidal ideation in those living alone.

	20–34 Years			35–64 Years			65–80 Years		
Model	PR	(95% CI)	EF (%)	PR	(95% CI)	EF (%)	PR	(95% CI)	EF (%)
**Depressive** **symptoms**									
Base (sex and age adjusted)	1.22	(0.92–1.62)		1.97	(1.69–2.29)		1.52	(1.31–1.75)	
+ SES total	1.38	(1.02–1.87)	−70%	1.39	(1.19–1.63)	60%	1.18	(1.02–1.37)	65%
+ education level	1.27	(0.96–1.69)	−23%	1.83	(1.57–2.12)	15%	1.48	(1.29–1.71)	7%
+ occupation	1.32	(1.00–1.76)	−45%	1.94	(1.67–2.26)	3%	1.52	(1.31–1.75)	0%
+ household income	1.18	(0.89–1.56)	21%	1.52	(1.30–1.77)	46%	1.27	(1.10–1.47)	47%
+ house owning	1.21	(0.90–1.64)	6%	1.72	(1.47–2.01)	26%	1.37	(1.18–1.59)	28%
+ residential area	1.23	(0.93–1.64)	−4%	1.96	(1.68–2.28)	1%	1.49	(1.29–1.72)	6%
+ Physical health	1.18	(0.89–1.56)	19%	1.60	(1.37–1.87)	38%	1.45	(1.25–1.66)	14%
+ SES total and physical health	1.37	(1.01–1.85)	−65%	1.27	(1.08–1.49)	73%	1.17	(1.01–1.36)	66%
**Suicidal ideation**									
Base (sex and age adjusted)	1.68	(1.10–2.57)		3.01	(2.44–3.73)		1.53	(1.21–1.93)	
+ SES total	1.81	(1.15–2.85)	−19%	1.77	(1.41–2.23)	62%	1.01	(0.81–1.27)	98%
+ education level	1.87	(1.23–2.85)	−28%	2.62	(2.11–3.25)	20%	1.48	(1.17–1.85)	10%
+ occupation	1.85	(1.20–2.85)	−25%	2.85	(2.31–3.52)	8%	1.51	(1.20–1.90)	3%
+ household income	1.49	(0.98–2.28)	28%	2.07	(1.66–2.59)	47%	1.18	(0.94–1.48)	67%
+ house owning	1.41	(0.89–2.22)	40%	2.35	(1.88–2.93)	33%	1.25	(1.00–1.58)	52%
+ residential area	1.71	(1.12–2.63)	−5%	2.98	(2.41–3.69)	2%	1.50	(1.19–1.89)	5%
+ Physical health	1.62	(1.06–2.47)	10%	2.17	(1.74–2.71)	42%	1.40	(1.12–1.74)	25%
+ SES total and physical health	1.82	(1.16–2.84)	−20%	1.55	(1.22–1.96)	73%	1.00	(0.80–1.25)	100%

SES, socioeconomic status; PR, prevalence ratio; CI, confidence intervals, EF, explained fraction. Physical health: number of chronic diseases and mobility, self-care, usual activities, and pain/discomfort by EQ-5L-3 level questionnaires. All PRs were those of living alone relative to those living with others for depressive symptoms or suicidal ideation. All were adjusted for variables listed in the corresponding row and additionally for sex and 5-year age group. EFs (%) was the proportion of excess prevalence explained by each variable or the combination of the variables, which were calculated as [(PR-1) − (PRa-1)]/(PR-1), where PR is that of the living alone from the base model and PRa is the PR of living alone in the model including the potential mediating or confounding variables in addition to the variables in the base model.

## Data Availability

All data were applicable from the following hyperlinks to publicity archived datasets analyzed; at https://knhanes.kdca.go.kr/knhanes/sub03/sub03_02_05.do (accessed on 26 October 2021).

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
