# Peer review of "Depressive Symptoms and Suicidal Ideation in Individuals Living Alone in South Korea"

_diagnostics, 2022, doi:10.3390/diagnostics12030603_

Round 1
Reviewer 1 Report
The study is well-conducted and presents a crucial mental health follow-up issue. Furthermore, it brings conclusions that are plausible and logical in view of the socio-economic context. The findings indicate that more lonely individuals were more likely to have depressive symptoms compared to those with a standard social environment. In view of the above, I consider the manuscript to be a favorable opinion. I congratulate the authors for the excellent conduct of the study.
Author Response
Thank you for your comments. We found that individuals living alone at all ages had higher rates of depressive symptoms and suicidal ideation than those living with others, and that SES and physical health contributed significantly to these mental health differences in 35-year- and older individuals living alone and those living with others. However, in the younger age group of 20-34 years, we found that SES and physical health did not significantly explain the differences in depressive symptoms and SI between individuals living alone and those living with others. Our study is the first to confirm that the factors that explain the difference in mental health between individuals living alone and those living with others differ by age group, and this is the strength of our study.
Reviewer 2 Report
Here are some concerns and questions need to be considered.
1. This study compared the prevalence of depressive symptoms/suicidal deation in individuals living alone and those living with others using datasets of KNHANES. However, no evidence supported the association of type of living(alone or not) on mental status in this datasets. Therefore, the multivariable logistic regression analysis of association of type of living
and mental status should be conducted first.
2. Line86: The nationwide cross-sectional survey was conducted every year. Why the author selected 2013, 2015, 2017, and 2019 data for analysis ? Explanation is needed.
3.Line146: why the authors divided all subjects into three group the 20–34, 35–64, and 65–80 years old ? Triple subjects number in 35-64 group may be responsible for these discrepancies. Four groups with 15-year interval may be more reasonable to analyze the differences in mental health according to age.
4.The authors need to illustrate the limitations. The most important limitation of the study is the cross-sectional design. Also, the data with one-time assessment were not as reliable as longitudinal data.
Reviewer 3 Report
The study peculiarities of Depressive symptoms and suicidal ideation in individuals living alone in South Korea. The main weakness of the study regards the novelty, I'm not sure what is being added novely to the literature.
There are already studies showing the relation between knowledge about this questions as (to name ones):
Suicidal ideation among the elderly living in the community: Correlation with living arrangement, subjective memory complaints, and depression
https://pubmed.ncbi.nlm.nih.gov/34710504/
Psychosocial risk profiles among older adults living alone in South Korea: A latent profile analysis
https://pubmed.ncbi.nlm.nih.gov/34004489/
Factors Influencing Suicidal Ideation and Attempts among Older Korean Adults: Focusing on Age Discrimination and Neglect
https://pubmed.ncbi.nlm.nih.gov/33672881/
Other issues:
Material and Methods: In this section, you need to clearly describe how individuals were approached, how many were approached, how many were eligible, consented or refused. Also, Inclusion and exclusion criteria should be cited with references and cited guidelines for cross sectional observational study may be recommended in order to improve the quality of the manuscript.
The Discussion section is a rehashing of the results. It does not appear that the authors include much interpretation of what the study findings mean for clinical practice or research.
FInally, the conclusión is weak and too long.
I consider that the study is not ready for publication and regret that the disposition is not favorable, but would like to thank you for your support.
We wish you all the best.
Round 2
Reviewer 3 Report
My opinion about the article remains the same of the first revision of manuscript. Most of the issues that I advanced to you cannot be repaired. A new study would be needed to make these things suitable. The clarifications provided do not solve the problem.